# Dissociation of two-dimensional excitons in monolayer WSe$_2$

Mathieu Massicotte[1], Fabien Vialla[1], Peter Schmidt[1], Mark B. Lundeberg[1], Simone Latini[2,3], Sten Haastrup[2], Mark Danovich [4], Diana Davydovskaya[1], Kenji Watanabe [5], Takashi Taniguchi[5], Vladimir I. Fal'ko[3], Kristian S. Thygesen [2,3], Thomas G. Pedersen [6,7] & Frank H.L. Koppens[1,8]

Two-dimensional (2D) semiconducting materials are promising building blocks for optoelectronic applications, many of which require efficient dissociation of excitons into free electrons and holes. However, the strongly bound excitons arising from the enhanced Coulomb interaction in these monolayers suppresses the creation of free carriers. Here, we identify the main exciton dissociation mechanism through time and spectrally resolved photocurrent measurements in a monolayer WSe$_2$ p–n junction. We find that under static in-plane electric field, excitons dissociate at a rate corresponding to the one predicted for tunnel ionization of 2D Wannier–Mott excitons. This study is essential for understanding the photoresponse of 2D semiconductors and offers design rules for the realization of efficient photodetectors, valley dependent optoelectronics, and novel quantum coherent phases.

[1] ICFO–Institut de Ciències Fotòniques, The Barcelona Institute of Science and Technology, Castelldefels, Barcelona 08860, Spain. [2] CAMD, Department of Physics, Technical University of Denmark, 2800 Kgs Lyngby, Denmark. [3] Center for Nanostructured Graphene (CNG), Technical University of Denmark, Kongens, Lyngby 2800, Denmark. [4] National Graphene Institute, University of Manchester, Booth St E, Manchester M13 9PL, UK. [5] National Institute for Materials Science, 1-1 Namiki, Tsukuba 305-0044, Japan. [6] Department of Physics and Nanotechnology, Aalborg University, DK-9220 Aalborg East, Denmark. [7] Center for Nanostructured Graphene (CNG), DK-9220 Aalborg Øst, Denmark. [8] ICREA – Institució Catalana de Recerça i Estudis Avancats, 08010 Barcelona, Spain. Correspondence and requests for materials should be addressed to F.H.L.K. (email: frank.koppens@icfo.eu)

As Johan Stark first observed in hydrogen atoms[1], applying an electric field on Coulomb-bound particles shifts their energy levels and eventually leads to their dissociation (Fig. 1a). In condensed matter physics, Wannier–Mott excitons display features analogous to those of hydrogen[2], but with the crucial difference that they recombine if they are not dissociated. Thermal energy is usually sufficient to ionize excitons in 3D semiconductors owing to their small binding energy $E_B$ (typically a few meV). In contrast, quantum confinement effects and reduced Coulomb screening in low-dimensional materials give rise to large exciton binding energy ($E_B > 100$ meV), which prevents thermal or spontaneous dissociation even at elevated temperatures and exciton densities.

In particular, monolayer transition metal dichalcogenides (TMDs) have aroused tremendous interest due to their unique optical properties governed by prominent excitonic features[3–6] and spin- and valley dependent effects[7–11]. These 2D semiconductors provide an exciting testbed for probing the physics arising from many-body Coulomb interactions[6,12]. Recently, all-optical experiments have revealed a wealth of physical phenomena such as exciton[13,14], trion[15,16], and biexciton[17] formation, bandgap renormalization[18], exciton–exciton annihilation[19–25], and optical Stark effect[7,11]. Exciton dissociation, on the other hand, can in principle be assessed through photocurrent measurements since photocurrent directly stems from the conversion of excitons into free carriers. A large number of studies have investigated photodetection performances of 2D TMDs[26–29] and demonstrated their potential as photodetectors and solar cells. However, it is still unclear which

dissociation process can overcome the large exciton binding energy and lead to efficient photocurrent generation in these devices. Theoretical studies suggest that strong electric fields may provide the energy required to dissociate the excitons[30–32], but the precise mechanism governing exciton dissociation in 2D TMDs remains to be experimentally investigated.

Here, we address this important issue by monitoring the exciton dissociation and subsequent transport of free carriers in a monolayer TMD p–n junction through spectrally and temporally resolved photocurrent measurements. Combining these two approaches allow us to assess and correlate two essential excitonic properties under static electric field, namely the Stark shift and the dissociation time. Further, we make use of the extreme thinness of 2D materials and their contamination-free assembly into heterostructures to reliably control the potential landscape experienced by the excitons. By placing the monolayer TMD in close proximity to metallic split gates, we can generate high in-plane electric fields and drive a photocurrent (PC). We find that at low field the photoresponse time of our device is limited by the rate at which excitons tunnel into the continuum through the potential barrier created by their binding energy, a process known as tunnel ionization (Fig. 1a). Tuning the electric field inside the p–n junction further allows us to disentangle various dynamical processes of excitons and free carriers and to identify the kinetic bottlenecks that govern the performance of TMD-based optoelectronic devices.

## Results

**Device structure and characterization.** Figure 1b, c presents a schematic and optical micrograph of our lateral p–n junction

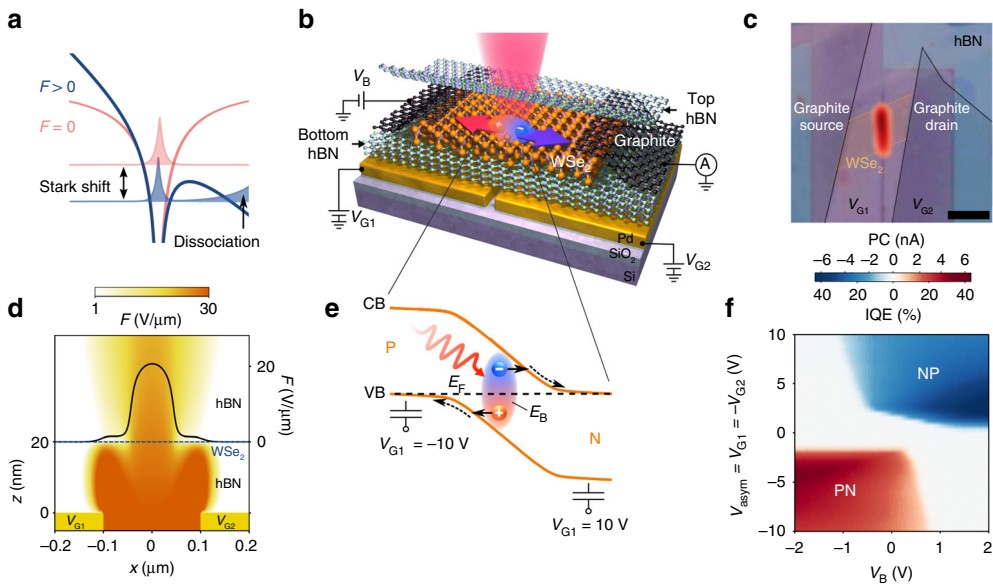

**Fig. 1** Photocurrent generation by exciton dissociation in a monolayer WSe₂ p–n junction. **a** Illustration of a Wannier–Mott exciton in the absence (red) and presence (blue) of a static electric field. The exciton wave functions are represented by the shaded curves while the exciton potentials are shown by the thick solid lines. **b** Schematic of a monolayer WSe₂ device controlled by two local metal gates with voltages $V_{G1}$ and $V_{G2}$. Two graphite flakes (colored in black) are placed on both sides of the WSe₂ layer (orange) and encapsulated between two hBN flakes (pale blue and green). **c** Optical image of a p–n junction device overlaid with a spatial PC map. The graphite and WSe₂ flakes are outlined and shaded for clarity. PC is measured at $V_{asym} = V_{G1} = -V_{G2} = -10$ V and $V_B = 0$ V, with a laser power $P = 1$ μW and a photon energy $h\nu = 1.65$ eV. The scale bar is 4 μm. **d** Side view of the electric field distribution $F$ across a device made of hBN (20 nm thick), monolayer WSe₂ (dotted blue line), and hBN (30 nm thick) atop metallic split gates separated by 200 nm (yellow rectangles). The field is calculated for $V_{asym} = -10$ V and $V_B = 0$. The color bar above indicates the magnitude of $\log_{10}(F)$. The in-plane electric field $F(x)$ inside the WSe₂ is shown by the solid black line (right axis). **e** Band diagram of the p–n junction between $x = -0.1$ and 0.1 μm (cf., Fig. 1d) calculated for $V_{asym} = -10$ V and $V_B = 0$ V. The solid orange lines represent the conduction (CB) and valence band (VB) and the black dotted line corresponds to the Fermi level ($E_F$). Following resonant optical excitation (red sinusoidal arrow), excitons are generated and dissociated via tunnel ionization (solid black arrows). The resulting free carriers drift out of the junction (dotted black arrows) and generate a photocurrent. **f** PC measured at the junction as a function of $V_{asym}$ and $V_B$, with a laser power $P = 0.5$ μW and a photon energy $h\nu = 1.65$ eV. The color bar between **c** and **f** displays the magnitude of the PC as well as the internal quantum efficiency, $\text{IQE} = \frac{PC}{e_0} \frac{h\nu}{AP}$, where $A = 5\%$ is the absorption coefficient

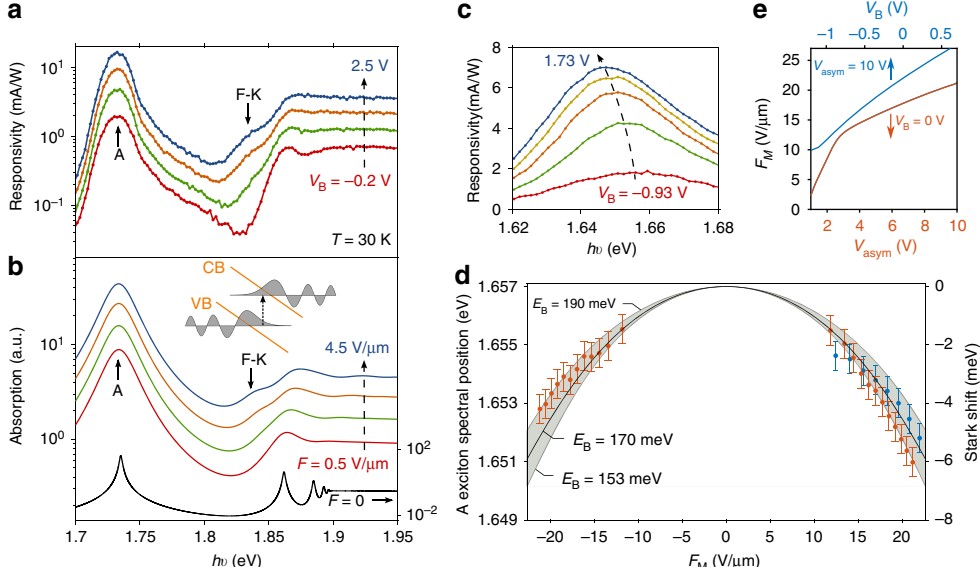

**Fig. 2** Electroabsorption and Stark effect in monolayer WSe$_2$ p–n junctions. **a** Responsivity (PC/P) spectra measured at the p–n junction of device 3 (Supplementary Fig. 2) for various $V_B$, with a laser power $P = 1\,\mu W$ and $T = 30\,K$. The spectra are vertically shifted (by a factor of 1.5) for clarity. **b** Absorption spectra calculated using a Wannier–Mott exciton model at different in-plane electric fields F. The solid colored lines (left axis) were calculated with a phenomenological line shape broadening of 15 meV, while the black solid line (right axis) was calculated without broadening. All spectra are vertically shifted for clarity. Inset: Schematics illustrating the sub-bandgap, field-induced absorption increase due to the Franz–Keldysh (F-K) effect. The application of an in-plane field tilts the conduction (CB) and valence band (VB) of the semiconductor (orange lines) and allows the wave functions of the free electrons and holes (gray shaded curves) to leak into the bandgap, which results in an increase of the sub-bandgap absorption (dotted arrow). **c** Responsivity spectra (around the A exciton) measured on device 1 (shown in Fig. 1c) at various $V_B$, with a laser power $P = 0.5\,\mu W$, $V_{asym} = 10\,V$ and at room temperature. **d** Position of the A exciton as a function of the calculated $F_M$ for the same values of $V_{asym}$ and $V_B$ as the one shown in **e**. Orange data points correspond to different values of $V_{asym}$ at $V_B = 0\,V$, while blue data points represent different values of $V_B$ at $V_{asym} = 10\,V$. Error bars correspond to the spectral resolution of our measurements. The calculated Stark shifts (right axis) induced by the $F_M$ are represented by the black solid line ($E_B = 170$ meV) and the gray shaded curves ($E_B = 153$ and $190$ meV). The latter values represent the uncertainty bounds of $E_B$ arising from the uncertainty (95% confidence interval) in the measured polarizability. **e** Maximum in-plane electric field $F_M$ calculated as a function of $V_{asym}$ (with $V_B = 0\,V$, solid orange line) and as a function of $V_B$ (with $V_{asym} = 10\,V$, solid blue line)

device made by assembling exfoliated flakes on metallic split gates ($V_{G1}$ and $V_{G2}$) separated by 200 nm (see "Methods"). Few-layer graphite flakes placed on both ends of a monolayer WSe$_2$ flake serve as ambipolar electrical contacts[33] that we use to apply a bias voltage $V_B$ and collect the photogenerated charges. The lateral graphite-WSe$_2$-graphite assembly is fully encapsulated in hexagonal boron nitride, typically 20 nm thick, which provides a clean and flat substrate. Three devices were measured (see Supplementary Note 1 and Supplementary Figs. 1–3), but unless otherwise specified, all measurements presented in the main text are obtained at room temperature from the device shown in Fig. 1c.

Tuning of bias and gate voltages allows us to finely control the in-plane electric field F. Finite-element and analytical calculations of the electric field distribution in our device (see Supplementary Note 2 and Supplementary Figs. 4–7) provide us with a precise estimate of F and the electrostatic doping inside the WSe$_2$ (Fig. 1d). Applying gate voltages of opposite polarity ($V_{asym} = V_{G1} = -V_{G2} = -10\,V$) leads to the formation of a sharp p–n junction (Fig. 1e) with an in-plane electric field reaching 21 V $\mu m^{-1}$ (Fig. 1d). The photoresponse that we observed at the junction (Fig. 1c) follows a photodiode-like behavior: PC is only generated in the p–n or n–p configuration (see Supplementary Fig. 1c) and can be increased by applying a reverse bias voltage (Fig. 1f).

**Spectral response.** We probe the absorption spectrum in the photoactive region by measuring the PC as a function of photon energy $h\upsilon$ at a constant laser power P and in-plane electric field F. Figure 2a shows the responsivity (PC/P) spectra of a device similar to the one presented in Fig. 1c, measured at various $V_B$ and at low temperature ($T = 30\,K$) in order to reduce thermal broadening. We observe a pronounced peak at a photon energy $h\upsilon = 1.73$ eV, corresponding to the A exciton, and a step-like increase around 1.87 eV. For increasing electric field, this step-like feature broadens and an additional shoulder appears at 1.83 eV.

To identify the various spectral features, we compare the experimental spectra with first-principles calculations for a monolayer WSe$_2$ embedded in hBN (see Supplementary Note 3 and Supplementary Fig. 8). By including the electronic screening from the hBN layers in the many-body $G_0W_0$ and Bethe–Salpeter Equation (BSE) frameworks[34] we obtain a bandgap of 1.85 eV and a lowest bound exciton at 1.67 eV in good agreement with the experimental spectra. To account for the effect of a constant in-plane electric field we use a model based on the 2D Wannier equation (see Supplementary Note 4 and Supplementary Fig. 9). In these model calculations, screening by the TMD itself as well as the surrounding dielectric materials is described via the Keldysh potential for the electron–hole interaction. Figure 2b shows calculated absorption spectra for different in-plane fields F. Excellent agreement between experiment and calculations is found assuming a bandgap of 1.9 eV, which yields a binding energy of $E_B = 170$ meV for the A excitons consistent with the first-principles calculations. The unbroadened spectrum calculated at zero field (Fig. 2b, solid black line) confirms the presence of multiple overlapping excited excitonic peaks below the

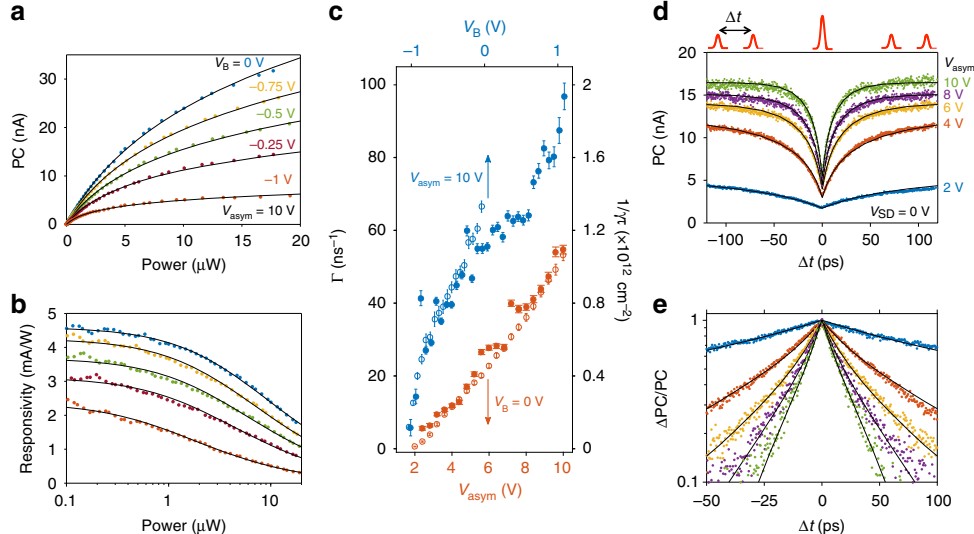

**Fig. 3** Determination of the photoresponse time $\tau$ by nonlinear and time-resolved photocurrent measurements. **a** PC vs. laser power $P$ for various $V_B$, at $V_{asym} = 10$ V and $h\nu = 1.65$ eV. **b** Responsivity (PC/P) in the same conditions as **a**. The solid black lines in **a** and **b** are fits to the data of PC $\propto \ln(1 + \gamma\tau N_0)$. **c** Photoresponse rate $\Gamma = \frac{1}{\tau}$ (filled circles, left axis) obtained from the TRPC measurements (shown in **d**, **e** and Supplementary Fig. 10d) and $\frac{\Gamma}{\gamma} = \frac{1}{\gamma\tau}$ (open circles, right axis) obtained from the power dependence measurements (shown in **a**, **b** and Supplementary Fig. 10a) as a function of $V_{asym}$ at $V_B = 0$ V (orange, lower axis) and $V_B$ at $V_{asym} = 10$ V (blue, top axis). Good agreement between TPRC and nonlinear PC measurements is found for an EEA rate of $\gamma = 0.05$ cm$^2$ s$^{-1}$. The error bars correspond to the standard deviations obtained from the fits. **d** PC as a function of time delay $\Delta t$ between two pulses (illustrated above the plot) at various value of $V_{asym}$, with time-averaged $P = 100$ μW and $V_B = 0$ V. **e** Same data as in **d** but plotted with the normalized $\frac{\Delta PC}{PC} = \frac{PC(\Delta t \rightarrow \infty) - PC(\Delta t)}{PC(\Delta t \rightarrow \infty) - PC(\Delta t = 0)}$. The solid black lines in **d** and **e** are fits to the data using the model described in the Supplementary Note 5

bandgap. The calculated spectra for higher field reproduce remarkably well the field-induced increase of the sub-bandgap absorption observed experimentally. This is a manifestation of the Franz–Keldysh effect, which results from the leakage of the free electron and hole wave functions into the bandgap (inset of Fig. 2b). We note that our experimental value of $E_B$ agrees well with the one estimated from the diamagnetic shift of a monolayer WSe$_2$ encapsulated between silica and hBN[35]. Larger $E_B$ has been observed in SiO$_2$-supported WSe$_2$ samples[36–38], underlining the role of the dielectric environment on the excitonic properties[39].

**Excitonic Stark effect**. Turning our attention to the A exciton photocurrent peak, we observe a pronounced red-shift as $V_B$ (Fig. 2c) and $V_{asym}$ increase. We attribute this to the DC Stark effect. In first approximation, the Stark shift of a 1s exciton (without dipole moment) is given by $\Delta E = -\frac{1}{2}\alpha F^2$, where $\alpha$ is the in-plane polarizability. As shown in Fig. 2d, the A exciton energy shows a quadratic dependence with the maximum in-plane electric field $F_M$ calculated for different values of $V_{asym}$ and $V_B$ (Fig. 2e), yielding a polarizability of $\alpha = (1 \pm 0.2) \times 10^{-6}$ Dm/V. This shift matches well with the predicted polarizability of $\alpha = 9.4 \times 10^{-7}$ Dm/V for $E_B = 170$ meV, thus supporting our previous spectral analysis. Interestingly, we note that the measured in-plane polarizability is two order of magnitude larger than the out-of-plane value recently obtained in PL experiments[40]. This strong anisotropy confirms the 2D nature of the A exciton and demonstrates the advantage of using in-plane electric fields for controlling the optical properties of TMDs[31].

**Photoresponse dynamics**. Along with the Stark shift, the application of a large in-plane electric field shortens the lifetime of excitons, which eventually decay into free electrons and holes (Fig. 1a). We probe these decay dynamics by assessing the photoresponse time $\tau$ of the device with time-resolved photocurrent measurements (TRPC), banking on the nonlinear photoresponse

of the WSe$_2$. Figure 3a, b shows the strong sublinear power dependence of the photocurrent (and the corresponding responsivity) under resonant pulsed optical excitation ($h\nu = 1.65$ eV, see "Methods"). Many physical processes may be responsible for or contribute to the observed sublinearity, including phase space filling[41] and dynamic screening effects (e.g., bandgap renormalization[18]). These many-body effects become intricate as the exciton gas approaches the Mott transition[42]. However, recent time-resolved spectroscopy[19,22] and photoluminescence[20,23] experiments indicate that in this exciton density regime ($10^{11} \lesssim N \lesssim 10^{13}$ cm$^{-2}$), exciton–exciton annihilation (EEA, or exciton Auger recombination) is the dominant decay process for excitons in TMDs[24]. To account for EEA in the rate equation governing the photocurrent we add a loss term that scales quadratically with the exciton density ($\gamma N^2$, where $\gamma$ is the EEA rate). Assuming that each pulse generates an initial exciton population $N_0$, this model yields PC $\propto \ln(1 + \gamma\tau N_0)$, which reproduces well the observed sublinear photoresponse (black lines in Fig. 3a, b, see Supplementary Note 5). Moreover, the fits capture adequately the variation of the sublinear photoresponse with bias (Fig. 3a, b) and gate (Supplementary Fig. 10a) voltages, from which we extract the values of $1/\gamma\tau$ (Fig. 3c). Hence, these nonlinear measurements already offer an indirect way to probe the photoresponse time.

In order to directly extract $\tau$, we resonantly excite A excitons in the $p$–$n$ junction with a pair of 200 fs-long laser pulses separated by a variable time delay $\Delta t$, for various values of $V_{asym}$ (Fig. 3d, e). Due to the sublinear power dependence, the photocurrent displays a symmetric dip when the two pulses coincide in time ($\Delta t = 0$). By extending our nonlinear photocurrent model to the case of two time-delayed pulses (see Supplementary Note 5 and Supplementary Fig. 10), we can show that the time dependence of this dip is dominated by an exponential time constant corresponding to the intrinsic photoresponse time $\tau$ of the device. The photoresponse rate $\Gamma = \frac{1}{\tau}$ is extracted from TRPC measurements at various values of $V_{asym}$ (Fig. 3d, e) and $V_B$ (see Supplementary Fig. 10d) and presented in Fig. 3c. We observe

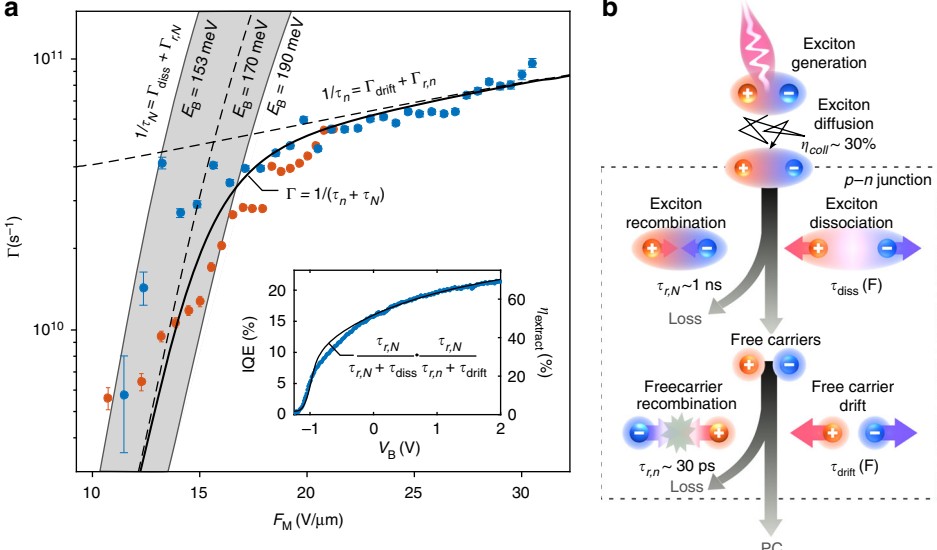

**Fig. 4** Dynamic processes governing the photoresponse of monolayer WSe$_2$ p–n junctions. **a** Photoresponse rate $\Gamma = \frac{1}{\tau}$ measured by TRPC (same data as Fig. 3c) vs. maximum in-plane electric field $F_M$ calculated for various values of $V_{asym}$ (with $V_B = 0$ V, orange points) and $V_B$ (with $V_{asym} = 10$ V, blue points). At low field ($F_M < 15$ Vμm$^{-1}$), the photoresponse is well described by the total exciton rate $\Gamma_N = \frac{1}{\tau_N} = \Gamma_{diss}(F_M) + \Gamma_{r,N}$, where $\tau_{r,N} = 1/\Gamma_{r,N}$ is the exciton lifetime at zero field ($\tau_{r,N} \sim 1$ ns) and $\Gamma_{diss}$ is the exciton dissociation rate predicted by the 2D Wannier–Mott exciton (see Supplementary Note 4) with a binding energy between $E_B = 153$ and 190 meV (gray shaded curves) and $E_B = 170$ meV (dotted black line). At high field ($F_M > 20$ V μm$^{-1}$), the photoresponse is governed by the total free carrier rate $\Gamma_n = \frac{1}{\tau_n} = \Gamma_{drift}(F_M) + \Gamma_{r,n}$ (dotted black line), where $\tau_{r,n} = 1/\Gamma_{r,n}$ is the free carrier lifetime at zero field ($\tau_{r,n} \sim 30$ ps) and $\Gamma_{drift}$ is the rate at which carriers (with a mobility $\mu = 4$ cm$^2$ V$^{-1}$ s$^{-1}$) drift out of the junction (see Supplementary Note 6). Since exciton dissociation and free carrier drift are consecutive processes, the total photoresponse rate of the device is $\Gamma \approx \frac{1}{\tau_N + \tau_n}$ (black solid line). Inset: IQE vs. $V_B$ measured at $V_{asym} = 10$ V extracted from Fig. 1f (left axis, blue data points). Extraction efficiency, $\eta_{extract} = \frac{\tau_{r,n}}{\tau_{r,n} + \tau_{drift}} \frac{\tau_{r,N}}{\tau_{r,N} + \tau_{diss}}$, calculated with our model vs. $V_B$ (right axis, black solid line). **b** Schematic of the processes contributing to the photoresponse of the device. Excitons are generated by resonant optical excitation and approximatively 30% ($\eta_{coll}$) of them reach the p–n junction by diffusion during their lifetime $\tau_{r,N}$. Excitons entering the p–n junctions (black dotted box) may either recombine with a time constant $\tau_{r,N}$ or dissociate by tunnel ionization at a rate $\Gamma_{diss}$. The resultant free carriers generate a photocurrent as they drift out of the junction at a rate $\Gamma_{drift}$, but a fraction is also lost due to their finite lifetime $\tau_{r,n}$. Holes and electrons are represented by red and blue spheres

that $\Gamma$ increases markedly with gate and bias voltages, and remarkably follows the same trend as the values of $1/\gamma\tau$ obtained from the power dependence measurements. Comparing these two results, we obtain an EEA rate of $\gamma = 0.05$ cm$^2$/s, which is similar to those found in WSe$_2$[19,23], MoS$_2$[21,22], and WS$_2$[20,25]. We also note that the shortest response time we measure, $\tau = 10.3 \pm 0.4$ ps, translates into a bandwidth of $f = 0.55/\tau \sim 50$ GHz, which compares with the fastest responses measured in TMD-based photodetectors[43,44].

## Discussion

To directly address the exciton dissociation caused by the in-plane electric field $F_M$, we examine the dependence of the photoresponse rate $\Gamma$ on $F_M$ at the p–n junction (Fig. 4a). Clearly, two regimes can be distinguished. The rapid increase of $\Gamma$ with $F_M$ is attributed to dissociation by tunnel ionization. We verify this by comparing the measured $\Gamma$ to the calculated tunnel ionization rate $\Gamma_{diss}$, obtained by introducing the complex scaling formalism in the 2D Wannier–Mott exciton model (see Supplementary Note 4 and Supplementary Table 1). According to this model, $\Gamma_{diss}$ can be evaluated in first approximation by the product of the "attempt frequency"[45], which scales with $E_B/h$, and the exponential tunneling term $\exp(-E_B/e_0 dF_M)$, where $e_0$ is the elementary charge, $d$ is the exciton diameter, and $h$ is the Plank constant. We find that the dependence of $\Gamma$ at low field ($F_M < 15$ V μm$^{-1}$) coincides well with the calculated dissociation rate of excitons with $E_B = 170$ meV, in agreement with our photocurrent spectroscopy analysis. More importantly, this shows that in the low-field

regime the exciton dissociation process is the rate-limiting step governing the generation of photocurrent. We note that in multilayer TMDs, where $E_B \sim 50$ meV, the ionization rate is two orders of magnitude larger than in the monolayer case[46], and hence this process was not found to limit the photoresponse rate of multilayer devices[44].

At high electric field ($F_M > 20$ V μm$^{-1}$), the photoresponse rate deviates from the dissociation rate-limited model and enters a new regime characterized by a more moderate increase of $\Gamma$ with $F_M$. The observed linear scaling of $\Gamma(F_M)$ suggests that, in this regime, the photoresponse rate is limited by the drift-diffusive transport of free carriers out of the p–n junction. By considering a carrier drift velocity $v_{drift} = \mu F$, we estimate that carriers generated in the center of the junction of length $L = 200$ nm escape the junction at a rate $\Gamma_{drift} = 2\mu F/L$. Comparing this simple expression (dotted line in Fig. 4a) to the measured $\Gamma$ at high field, we find $\mu = 4 \pm 1$ cm$^2$ V$^{-1}$ s$^{-1}$, which is very similar to the room temperature field-effect mobility that we measure in our sample ($\mu_{FE} \sim 3$ cm$^2$ V$^{-1}$ s$^{-1}$, see Supplementary Note 1).

A complete photocurrent model is achieved by introducing competing loss mechanisms caused by the radiative and non-radiative recombination of excitons (see Supplementary Note 6). Good agreement with the experimental data is obtained by considering the finite lifetime of excitons ($\tau_{r,N} = 1/\Gamma_{r,N} \sim 1$ ns[20,23], see Supplementary Note 1) and free carriers ($\tau_{r,n} = 1/\Gamma_{r,n} \sim 30$ ps[41]) at zero electric field. This comprehensive picture of the dynamical processes (Fig. 4b) offers valuable insights into the internal quantum efficiency (IQE) of the photocurrent generation mechanism in this device. Indeed, the efficiency $\eta$ of each

photocurrent step depends on the competition between the PC-generating ($\tau_{\text{drift}}$, $\tau_{\text{diss}}$) and the loss ($\tau_{\text{r},N/n}$) pathways, such that $\eta_{\text{diss/drift}} = \tau_{\text{r},N/n}/(\tau_{\text{r},N/n} + \tau_{\text{diss/drift}})$. In the inset of Fig. 4a, we compare the IQE measured at low power as a function of $V_{\text{B}}$ with the total extraction efficiency $\eta_{\text{extract}} = \eta_{\text{drift}} \eta_{\text{diss}}$ derived from the kinetic model shown in Fig. 4b. We find that $\eta_{\text{extract}}$ captures very well the bias dependence of the IQE, indicating that we correctly identified the relevant PC-generating processes. The field-independent discrepancy of 30% is attributed to the collection efficiency $\eta_{\text{coll}}$, which we define as the ratio between the number of excitons reaching the $p$–$n$ junction and the number of absorbed photons. This value coincides with our analysis of the measured photocurrent profile and with the prediction of our exciton diffusion model (see Supplementary Note 7 and Supplementary Fig. 11).

In summary, our study offers a global understanding of the fundamental mechanisms governing the exciton dynamics and associated photoresponse in monolayer TMDs under in-plane electric field. We demonstrate that despite their large binding energy, photogenerated excitons can rapidly dissociate into free carriers via tunnel ionization, thereby outcompeting recombination processes. Importantly, this knowledge allows us to identify the main material properties that limit photocurrent generation in TMDs such as carrier mobility, exciton binding energy, and lifetime. This provides guidelines in terms of device design, material quality improvement, and Coulomb engineering of the van der Waals heterostructure to further improve the performances of TMD-based optoelectronics devices and develop their applications in valleytronics. We finally note that the observed Stark and Franz–Keldysh effects open up exciting opportunities for modulating light with 2D materials[47].

## Methods

**Device fabrication**. Exfoliated layers are assembled in a van der Waals heterostructure using the same technique as described in ref. [48]. The monolayer of WSe$_2$ is identified by photoluminescence measurement (see Supplementary Note 1). The heterostructure is deposited onto metallic split gates (15 nm palladium) defined by electron-beam lithography on a degenerately doped silicon substrate covered with a 285-nm-thick SiO$_2$ layer. The two graphite flakes are electrically connected by one-dimensional contacts[48] made of Ti/Au (2/100 nm).

**Photocurrent measurements**. Photocurrent measurements are performed using a photocurrent scanning microscope setup, where a laser beam is focused by a microscope objective (Olympus LUCPlanFLN × 40) onto the device placed on a piezoelectric stage (Attocube ANC300). Photocurrent is measured with a pre-amplifier and a lock-in amplifier synchronized with a mechanical chopper. A supercontinuum laser (NKT Photonics SuperK Extreme), with a pulse duration of ~40 ps, repetition rate of 40 MHz and tunable wavelength (from 500 to 1500 nm) is employed to characterize the devices, perform photocurrent spectroscopy, and measure the photocurrent power dependence. Time-resolved photocurrent measurements are performed using a Ti:sapphire laser (Thorlabs Octavius) with ~200 fs pulses (at the sample), with a repetition rate of 85 MHz, and centered at $h\nu$ = 1.65 eV (FWHM = 0.07 eV), which corresponds to the A exciton absorption peak. The laser beam is split into two arms and recombined using 50/50 beamsplitters. A mechanical chopper modulates the laser beam in one arm (pump), while the other arm (probe) has a motorized translation stage that allows for the generation of a computer-controlled time delay $\Delta t$ between the two pulses.

**Data availability**. The data that support the findings of this study are available from the corresponding author on request.

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

## Acknowledgements

T.G.P. and K.S.T. acknowledge support for CNG by the Danish National Research Foundation, project DNRF103. T.P.G. also acknowledges support for the VKR center of excellence QUSCOPE by the Villum foundation. M.M. thanks the Natural Sciences and Engineering Research Council of Canada (PGSD3-426325-2012). P.S. acknowledges financial support by a scholarship from the "la Caixa" Banking Foundation. F.V. acknowledges financial support from Marie-Curie International Fellowship COFUND and ICFOnest program. F.H.L.K. acknowledges financial support from the Government of Catalonia trough the SGR grant (2014-SGR-1535), and from the Spanish Ministry of Economy and Competitiveness, through the "Severo Ochoa" Programme for Centres of Excellence in R&D (SEV-2015-0522), support by Fundacio Cellex Barcelona, CERCA Programme/Generalitat de Catalunya and the Mineco grants Ramón y Cajal (RYC-2012-12281) and Plan Nacional (FIS2013-47161-P and FIS2014-59639-JIN). Furthermore, the research leading to these results has received funding from the European Union Seventh Framework Programme under grant agreement no. 696656 Graphene Flagship and the ERC starting grant (307806, CarbonLight).

## Author contributions

M.M. conceived and designed the experiments under the supervision of F.H.L.K., M.M., D.D., and F.V. fabricated the samples. M.M. and F.V. carried out the experiments. M.M. performed the data analysis and discussed the results with F.H.L.K, F.V., and P.S. T.G.P. developed the Wannier–Mott exciton model. T.P.G, M.B.L., M.D., and V.I.F. performed the electrostatic calculations, and S.H., S.L., and K.S.T. performed the ab-initio calculations. K.W. and T.T. provided hBN crystals. M.M., F.V., P.S., and F.H.L.K. co-wrote the manuscript, with the participation of T.G.P. and K.S.T.

## Additional information

**Competing interests:** The authors declare no competing interests.

