## [Peer Review File · Nature Communications]

Reviewers' comments:

Reviewer #1 (Remarks to the Author):

The authors present detailed photocurrent measurements of monolayer WSe₂ encapsulated in hBN. They identify a peak in the photocurrent excitation spectrum as the onset of continuum states that agrees with BSE calculations of the exciton binding energy. They observe a DC Stark shift of the photocurrent excitation spectrum peak corresponding to the A-exciton and from it estimate the exciton polarizability. Using time-resolved photocurrent measurements they estimate an exciton-exciton annihilation rate of 0.05 cm²/s and a EEA-limited exciton lifetime of 10ps. The authors use a dissociation and drift model to describe the in-plane field dependence of the photocurrent and show that dissociation at low fields is consistent with tunnel ionization.

These results are novel, important, and of broad interest to the multidisciplinary field of researchers studying 2D materials. I think that the authors need to address a few questions in order to make the conclusions more convincing before the manuscript can be considered for publication.

1 - (a) Figure S2 e and f show significant difference in the photocurrent spectral responsivity of two different WSe₂ devices. However, the comparison is complicated by the fact that the measurements were not only on different samples but at vastly different temperatures. Could the authors clarify if the changes in the responsivity arise from temperature changes or from sample-to-sample variations?

(b) Further, these differences should give rise to uncertainty in the exciton binding energy, though none is given in the text. Please provide an estimate of the uncertainty in the binding energy, since it is integral in much of the subsequent analysis.

2- At $N > 11 \text{ cm}^{-2}$, the exciton binding energy will be further reduced by dynamic screening of the Coulombic interactions by the photoinjected excitons and charge. Though this process has not yet been directly measured, a number of theoretical studies (e.g. Nano Lett 16 5568 '16) have outlined the effects of dynamic screening and its effect on the exciton resonance is well known. Fundamentally, this is the process responsible for photo-induced bandgap renormalization and the insulator-metal transition described in ref 18 (Nat. Photon 9 466 '15). The authors appear not to have considered the effect of dynamic screening on the binding energy in their analysis. Depending on the absorbed fluence (i.e. excitation density) used in each measurement, the binding energy may be reduced well below the 170meV value considered here. This reduction may play an important role in the rapid ionization of excitons reported here.

3 - Chernkov et al PRL 115 126802 '15 report on electrical tuning of the exciton binding energy in WS₂. In that work, an applied gate voltage is found to inject charge, which screens the Coulombic interaction and thereby reduced the exciton binding energy. Have the authors considered the possibility of charge injection due to the applied voltage?

4- The authors study exciton dissociation in WSe₂ encapsulated in hBN. The increased dielectric contrast of hBN screens the Coulombic interaction and thereby reduces the exciton binding energy, as noted by the authors. Based on the experimental and theoretical results presented here, can the authors comment on whether tunnel ionization will be dominant in free-standing WSe₂ where the binding energy is considerably higher?

5- The mobility of 4 cm²/Vs estimated on page 7 is much smaller than the ~100 cm²/Vs value in literature Nat. Nano. 7 699 '12. Is this value limited by defects? How much does it vary among the three devices in this study?

6- The EEA work referenced by the authors report EEA rates that are an order of magnitude higher than those reported here. See Ref 19 and 21. The authors may not be aware of more

recent work (PR B 93 201111R '16, JPC Lett 7 5242 '16) that is in closer agreement with the estimates given here. Similar EEA-limited lifetimes of ~ 10 ps are given in those works.

Reviewer #2 (Remarks to the Author):

In, dissociation of 2D excitons in monolayer WSe₂, the authors perform optoelectronic measurements on WSe₂ electrostatically defined PN junctions. They study how an in-plane electric field leads to dissociation of the excitons as determined by two-pulse photo current measurements. Overall, I believe the paper is suitable for publication in Nature Communications and of general interest, but several points need to be clarified (below).

- 1) In the $\exp(-EB/e0dF)$ on page 6, do the authors mean F or Fm?
- 2) I do not understand exactly what the authors mean by, "According to this model, Γ_{diss} can be evaluated in first approximation by the product of the uncertainty-limited exciton lifetime EB/\hbar " This should be clarified.
- 3) Figure 2d, y axis should read -exciton energy or spectral position to avoid.
- 4) In the explanation of Figure 4, the authors discuss 2 regimes that are qualitatively obvious from Figure 4a, above and below 20 V/micron, which I believe they compare to the field associated with the exciton binding energy. For clarify, the authors should compare either two electric fields directly.

Reviewer #3 (Remarks to the Author):

The authors report a photoresponse study of a monolayer WSe₂ p-n junction device using spectrally- and time-resolved photocurrent measurements. They establish tunnel ionization as the major exciton dissociation mechanism through model fitting and show that the photoresponse rate is dissociation-limited below ~ 35 GHz and becomes drift velocity-limited above ~ 35 GHz. Their results provide direct comparison between the experiments and model, which are valuable in the study of TMD photo detectors. However, it is not clear that this work has sufficient novelty and significance for the readers, given that similar device geometry, comparable device performance, and modeling are available in literature. In addition, some of the central analysis appear insufficiently supported. Due to the above reasons, I would not recommend this manuscript for publication in Nature Communication unless the authors can properly address these issues.

More specific comments are listed below.

1. The authors use an uncertainly-limited recombination time in the tunnel ionization model (Page 6) and a recombination time of ~ 1 ns from literatures in the discussion of the relevant dynamic processes (Page 7). These numbers are not completely reliable and can be an overestimation. It is well known that the recombination in current thin-layered TMDs is mostly dominated by non-radiative processes, leading to a large variation of the reported recombination time from below ps to 1 ns at room temperature (10.1364/JOSAB.33.000C39). In other words, the recombination time in thin TMDs is often sample-dependent and needs to be measured from sample to sample for a reliable value. Similar issue could arise for lifetime of free carriers due to defect-induced carrier trapping (Page 7). As the relevant dynamics is a central idea in this work, the authors should either perform measurements to directly access the recombination and free carrier dynamics or provide arguments to justify the lifetimes they use in the analysis.

2. Fig. 4, the authors claim that the dissociation process is the rate-limiting factor for electric field of 10 – 15 V/ μm and the response rate matches that predicted by the tunnel ionization model. These statements are the main finding of this work but appear weakly supported. The model curve in Fig 4, which is the only evidence for the tunnel ionization process, was not directly generated from the model but rather extrapolation using the computed values at larger fields of 18- 24 V/ μm with basically four parameters (Table S1). It is understandable that there are difficulties and limitation in obtaining model curves at small fields, but this indirect approach unavoidably weakens the reliability of their analysis. The authors can consider comparing their measured Stark shift to that predicted by their tunnel ionization model as a possible further support or provide other evidence to back up this main claim.

3. A similar exciton ionization model has been reported recently (New J. Phys. 18 (2016) 073043), which does not rely on an extrapolation at low electric fields and predicts an ionization rate that is more than two-order-of-magnitude larger than the results in this manuscript. Given the direct relevance, this work should be cited and the authors should consider explanations for the above discrepancy.

4. In the analysis, the authors do not consider carrier travel time from the junction to the graphite source/drain. A quick estimate using the mobility of $4 \text{ cm}^2/\text{Vs}$, travel distance of $2 \mu\text{m}$, and bias voltage of 1 V yields a drift velocity $v_d = \mu E = 200 \text{ m/s}$ and a corresponding travel time of 10 ns, translating to 0.1 GHz. This is much slower than the extracted response rate of up to 50 GHz. Can the authors comment on this?

5. On a similar note to the previous comment, fast response rate of up to 60 GHz is reported at $V_B = 0$ (Fig. 3c) where the collection of photocurrent relies on the diffusion of carriers to the source and drain on μm scales. Is the carrier diffusion fast enough to yield such fast response rate?

6. The authors state that “the application of a large in-plane electric field shortens the lifetime of excitons” (Page 5). It is known that an external electric field reduces the overlap of electron and hole wave function and decreases exciton lifetime in the out-of-plane geometry. I would expect similar effect to take place in in-plane geometry. Can the authors clarify this?

7. What is the typical dark current range in the WSe₂ device? In Fig. 3a, it looks like the dark current in that specific gating condition is well below nA range. It will be informative if the authors can include IV characteristics at a few representative gate conditions with and without illumination.

8. The authors mention that the in-plane polarizability is much greater than the out-of-plane one in atomically-thin TMDs. This is an interesting observation. Can the author provide a short explanation for it?

We thank all referees for their constructive and generally positive comments on our manuscript.

We respond to the reviewers' comments below.

Please note that reviewers' comments are in blue font and that all changes made to the manuscript and Supplementary Information (SI) are highlighted in yellow.

Reviewer #1 (Remarks to the Author):

The authors present detailed photocurrent measurements of monolayer WSe₂ encapsulated in hBN. They identify a peak in the photocurrent excitation spectrum as the onset of continuum states that agrees with BSE calculations of the exciton binding energy. They observe a DC Stark shift of the photocurrent excitation spectrum peak corresponding to the A-exciton and from it estimate the exciton polarizability. Using time-resolved photocurrent measurements they estimate an exciton-exciton annihilation rate of 0.05 cm²/s and a EEA-limited exciton lifetime of 10ps. The authors use a dissociation and drift model to describe the in-plane field dependence of the photocurrent and show that dissociation at low fields is consistent with tunnel ionization.

These results are novel, important, and of broad interest to the multidisciplinary field of researchers studying 2D materials. I think that the authors need to address a few questions in order to make the conclusions more convincing before the manuscript can be considered for publication.

1 - (a) Figure S2 e and f show significant difference in the photocurrent spectral responsivity of two different WSe₂ devices. However, the comparison is complicated by the fact that the measurements were not only on different samples but at vastly different temperatures. Could the authors clarify if the changes in the responsivity arise from temperature changes or from sample-to-sample variations?

First, we thank the reviewer for his/her thoughtful and valuable feedback. There are indeed differences between photocurrent spectral response of these two devices (2 and 3) which, as the reviewer correctly points out, arise from (i) temperature changes and (ii) sample-to-sample variations.

i) Temperature changes account for the main spectral differences: the spectrum measured at low temperature (30 K) displays sharper feature (due to reduced thermal broadening) and is blue-shifted compared to the spectrum measured at room temperature. This shift, often observed in semiconductors, is due to an increase of the WSe₂ bandgap with temperature and is identical to the one reported by Arora et al. (Nanoscale 7 10421 '15).

ii) We also observe a relatively small change (less than a factor 2) in the maximum magnitude of the responsivity. Although we cannot exclude the effect of temperature, our measurements suggest that this change in magnitude is due to sample-to-sample variation. We indeed observe a similar change in the responsivity magnitude (less than a factor 2) between devices 1 and 2, which were both measured at the same temperature (300 K).

To address this comment, we added some comments in Section S1

(b) Further, these differences should give rise to uncertainty in the exciton binding energy, though none is given in the text. Please provide an estimate of the uncertainty in the binding energy, since it is integral in much of the subsequent analysis.

i) We do not expect the exciton binding energy to change with temperature. As Arora et al. point out, the fact that the exciton resonance energies follow the change of the bandgap with temperature “suggests the independence of the exciton binding energy on temperature”.

ii) However, variation in device geometry may affect the dielectric environment of the exciton and thus, its binding energy. To avoid this undesirable effect, all devices were made using similar heterostructures: a monolayer of WSe₂ encapsulated between two thick (> 20 nm) hBN layers. Since the thickness of the hBN is much larger than the exciton Bohr radius (~ 1 nm), we expect the dielectric environment of the exciton, and therefore its binding energy, to be similar.

In our study, the experimental uncertainty in the binding energy arises from the polarizability value α which we obtain by fitting the Stark shift. This fit yields a polarizability of $(1.0 \pm 0.2) \times 10^{-6}$ Dm/V. The uncertainty, slightly larger than the one reported in the previous manuscript, represents 95% confidence interval. To translate this uncertainty in α into an uncertainty in binding energy, we calculate the polarizability of excitons for a range of binding energies using the Wannier-Mott model presented in section 3 of the SI. We find that the maximum (1.2×10^{-6}) and minimum (0.8×10^{-6}) polarizability correspond to binding energies of 153 and 190 meV, respectively. The Stark shift predicted for these binding energies are shown in Fig. 2d. A sentence has been added to the figure caption to clarify this point.

2- At $N > 11 \text{ cm}^{-2}$, the exciton binding energy will be further reduced by dynamic screening of the Coulombic interactions by the photoinjected excitons and charge. Though this process has not yet been directly measured, a number of theoretical studies (e.g. Nano Lett 16 5568 '16) have outlined the effects of dynamic screening and its effect on the exciton resonance is well known. Fundamentally, this is the process responsible for photo-induced bandgap renormalization and the insulator-metal transition described in ref 18 (Nat. Photon 9 466 '15). The authors appear not to have considered the effect of dynamic screening on the binding energy in their analysis. Depending on the absorbed fluence (i.e. excitation density) used in each measurement, the binding energy may be reduced well below the 170meV value considered here. This reduction may play an important role in the rapid ionization of excitons reported here.

As the reviewer correctly points out, the dynamic screening of the Coulomb interaction can strongly affect the electronic and optical properties of TMDs, as evidenced by the observation of photo-induced bandgap renormalization. In our previous manuscript, we also acknowledged that “many physical processes may be responsible for or contribute to the observed sublinearity, including phase space filling and bandgap renormalization”. (For the sake of clarity, we now mention “dynamic screening” in the manuscript.) We considered the potential implications of dynamic screening in our study, however there are several reasons why we did not include it in our analysis:

- 1) The dynamic screening effects described in ref 18 (Nat. Photon 9 466 '15) have only been observed at high excitation density ($N > 1 \text{e}13 \text{ cm}^{-2}$) and become dominant at $1 \text{e}14 \text{ cm}^{-2}$. In

our study, the excitation density is always $N < 5 \times 10^{12} \text{ cm}^{-2}$.

- 2) The model that we use is based on exciton-exciton annihilation which has been shown to dominate at ‘intermediate’ excitation densities: $1 \times 10^{11} < N < 1 \times 10^{13} \text{ cm}^{-2}$ (n.b., we added this upper bound to the new manuscript). This relatively simple model (explained in more details in Section 5 of the SI) reproduces well the time-resolved photocurrent measurements performed at various laser fluences (corresponding to $1 \times 10^{11} < N < 5 \times 10^{12} \text{ cm}^{-2}$). Hence, dynamic screening effects appears to be negligible.
- 3) Finally, the expected effects caused by dynamic screening are **not** consistent with our experimental observations. Indeed, as the reviewer suggests, dynamic screening would lead to a reduction of the exciton binding energy. This would then increase the exciton ionization rate and therefore the magnitude of the measured photocurrent. Thus, we would expect the photocurrent to increase superlinearly with laser power (or excitation density). We however observe the opposite: a sublinear power dependence of the photocurrent (Fig. 3a).

3 - Chernkov et al PRL 115 126802 '15 report on electrical tuning of the exciton binding energy in WS₂. In that work, an applied gate voltage is found to inject charge, which screens the Coulombic interaction and thereby reduced the exciton binding energy. Have the authors considered the possibility of charge injection due to the applied voltage?

As the reviewer mentions, Coulomb screening caused by gate-induced charges (i.e. electrostatic doping) has been shown to reduce the exciton binding energy E_B in TMDs. While this effect probably takes place in the electrostatically doped regions of our device, it does not occur in the region between two gates where the in-plane electric field F , and therefore ionization, are highest.

To support this point, we calculated the charge density distribution n (using the numerical method presented in Section 2.1 of the SI) across the p-n junction (right axis of Fig. R1 shown below). We compared this charge distribution to the in-plane field in the same condition (left axis of Fig. R1 shown below) and see, indeed, that the field is highest in the charge neutral (undoped) zone. Hence, the exciton ionization rate (determined by the ratio E_B/F) is always higher in this region where the exciton binding energy is unaffected by Coulomb screening. This means that the exciton binding energy relevant to our analysis is the one corresponding to the undoped WSe₂. We included the above discussion and Fig. R1 to Section S2 of the SI.

We also performed more detailed calculations (not shown) of E_B in doped WSe₂ using the Wannier-Mott model (presented in Section 3 of the SI) which confirm that the ionization rate is largest in the undoped, inter-gate region.

Figure R1. Spatial distribution of the in-plane electric field F (left axis, blue curve) and charge carrier density n (right axis, red curve) inside the WSe₂ layer across the p-n junction. The device geometry and voltages applied are the same as those of Fig. 1d of the manuscript.

4- The authors study exciton dissociation in WSe₂ encapsulated in hBN. The increased dielectric contrast of hBN screens the Coulombic interaction and thereby reduces the exciton binding energy, as noted by the authors. Based on the experimental and theoretical results presented here, can the authors comment on whether tunnel ionization will be dominant in free-standing WSe₂ where the binding energy is considerably higher?

Indeed, the binding energy is considerably higher in free-standing WSe₂, so for a given in-plane electric field, the ionization rate should be significantly lower than in our hBN-encapsulated samples. Using the same Wannier-Mott model as presented in Section 3 of the SI, we calculated an exciton binding energy of 0.5 eV for free-standing WSe₂. The predicted Stark shift and ionization rates for different dielectric environment (κ) are shown in Fig. R2, which is included in the new version of the SI. For free-standing WSe₂ ($\kappa = 1$), these calculations indicate that the in-plane electric field must be higher than 30 V/micron in order for tunnel ionization to outcompete exciton recombination (assuming an exciton recombination time of 1 ns) and thus generate a photocurrent.

Figure R2. Stark shift and tunnel ionization rate vs in-plane electric field for WSe₂ surrounded by different dielectric environments with dielectric constants κ .

5- The mobility of $4 \text{ cm}^2/\text{Vs}$ estimated on page 7 is much smaller than the $\sim 100 \text{ cm}^2/\text{Vs}$ value in literature Nat. Nano. 7 699 '12. Is this value limited by defects? How much does it vary among the three devices in this study?

As we mention in the SI (p. 2) a wide range of mobilities have been reported in the literature, from 0.1 to $100 \text{ cm}^2/\text{Vs}$ (the latter being more the exception than the rule). Understanding the factors limiting the mobility of a particular device requires an in-depth analysis of transport measurements (typically using a 4-probe configuration), which fall outside the scope of our study.

The 2 contacts on our device allows us to only estimate its mobility, which, as we mention in the manuscript, agrees well with the mobility we obtained from the analysis of the time-resolved photocurrent measurements. While the field-effect mobility of the other devices was not measured, time-resolved photocurrent measurements yield similar mobilities.

6- The EEA work referenced by the authors report EEA rates that are and order of magnitude higher than those reported here. See Ref 19 and 21. The authors may not be aware of more recent work (PR B 93 201111R '16, JPC Lett 7 5242 '16) that is in closer agreement with the estimates given here. Similar EEA-limited lifetimes of $\sim 10\text{ps}$ are given in those works.

We thank the reviewer for drawing our attention to these articles which we included as references in the new version of our manuscript.

Reviewer #2 (Remarks to the Author):

In, dissociation of 2D excitons in monolayer WSe₂, the authors perform optoelectronic measurements on WSe₂ electrostatically defined PN junctions. They study how an in-plane electric field leads to dissociation of the excitons as determined by two-pulse photo current measurements. Overall, I believe the paper is suitable for publication in Nature Communications and of general interest, but several points need to be clarified (below).

1) In the $\exp(-EB/e_0dF)$ on page 6, do the authors mean F or F_M ?

First, we thank reviewer for his/her positive review and useful suggestions. The expression $\exp(-EB/e_0dF)$ corresponds to a more general situation. In our case, we indeed calculate Γ_{diss} using F_M . For sake of clarity, we replaced F by F_M in the manuscript.

2) I do not understand exactly what the authors mean by, "According to this model, Γ_{diss} can be evaluated in first approximation by the product of the uncertainty-limited exciton lifetime EB/\hbar " This should be clarified.

We agree that a more detailed explanation is needed to understand the origin of the EB/\hbar . This term represents, in first approximation, the frequency at which the exciton "attempts" to tunnel. A nice explanation of this term is given by Perebeinos et al. (Nano Lett. 7 609 '07):

“To convert the tunneling probability into the decay rate, we need to know the “attempt” frequency for tunneling. From the Heisenberg uncertainty principle, an averaged momentum of the bound exciton is $p = 2\pi\hbar/\lambda$, where λ is the exciton size. According to the virial theorem, the exciton binding energy E_B is proportional to the kinetic energy $E_K \approx pv/2$, where v is an averaged velocity. In the 3D case, $E_B = E_K$, and therefore one would expect in the case of nanotubes an attempt frequency $v/(2\lambda)$ to scale with $E_B/(2\pi\hbar)$. “

We rewrote this part of the manuscript to clarify this point and refer to Perebeinos et al.

3) Figure 2d, y axis should read –exciton energy or spectral position to avoid.

We thank the reviewer for his suggestion which we implemented in the new version of the manuscript.

4) In the explanation of Figure 4, the authors discuss 2 regimes that are qualitatively obvious from Figure 4a, above and below 20 V/micron, which I believe they compare to the field associated with the exciton binding energy. For clarify, the authors should compare either two electric fields directly.

The electric field threshold that the reviewer mentions (~20 V/micron) is determined not only by the exciton binding energy (and its associated field) but also by other parameters of our model (charge carrier mobility and recombination time). The comparison of this threshold field and the exciton field is therefore irrelevant.

Reviewer #3 (Remarks to the Author):

The authors report a photoresponse study of a monolayer WSe2 p-n junction device using spectrally- and time-resolved photocurrent measurements. They establish tunnel ionization as the major exciton dissociation mechanism through model fitting and show that the photoresponse rate is dissociation-limited below ~35 GHz and becomes drift velocity-limited above ~35 GHz. Their results provide direct comparison between the experiments and model, which are valuable in the study of TMD photo detectors. However, it is not clear that this work has sufficient novelty and significance for the readers, given that similar device geometry, comparable device performance, and modeling are available in literature. In addition, some of the central analysis appear insufficiently supported. Due to the above reasons, I would not recommend this manuscript for publication in Nature Communication unless the authors can properly address these issues.

We thank the reviewer for his/her constructive feedback and for giving us the opportunity to address the issues that he/she raised. However, we strongly disagree with the reviewer’s statement about the possible lack of novelty of our work. First of all, the referee did not mention any specific references to sustain this claim. While we do acknowledge (and cite) works focusing on the performances of similar devices, we stress that the main goal (and achievement) of our study is not to replicate these performances but to understand their **physical origin**, and in particular the exciton dissociation mechanism. Understanding the latter is crucial for the development of optoelectronic applications based on 2D materials and therefore requires in-depth experimental investigation. For this reason, we believe that our work, which addresses for the first time the exciton dissociation mechanism in 2D materials, is not only novel and original, but also of paramount importance to the multidisciplinary researchers studying 2D materials.

More specific comments are listed below.

1. The authors use an uncertainly-limited recombination time in the tunnel ionization model (Page 6) and a recombination time of ~ 1 ns from literatures in the discussion of the relevant dynamic processes (Page 7). These numbers are not completely reliable and can be an overestimation. It is well known that the recombination in current thin-layered TMDs is mostly dominated by non-radiative processes, leading to a large variation of the reported recombination time from below ps to 1 ns at room temperature (10.1364/JOSAB.33.000C39). In other words, the recombination time in thin TMDs is often sample-dependent and needs to be measured from sample to sample for a reliable value. Similar issue could arise for lifetime of free carriers due to defect-induced carrier trapping (Page 7). As the relevant dynamics is a central idea in this work, the authors should either perform measurements to directly access the recombination and free carrier dynamics or provide arguments to justify the lifetimes they use in the analysis.

We agree that the exciton and free carrier recombination times are often sample-dependent and that there is a large variation among the reported values. We however note that in our work, 1) the values are obtained by fitting our model to the measured photoresponse rate and IQE, not simply taken at random from the literature, and 2) the main results on which we report (exciton dissociation, carrier drift, IQE) are not significantly affected by those lifetime values.

To support these points, we recently measured the exciton recombination time on a similar sample (hBN-encapsulated WSe_2) through time-resolved PL measurements and extracted an exciton recombination time of ~ 300 ps (see Fig R3), in reasonable agreement with the value extracted from Fig 4a of our manuscript (~ 1 ns). In figure R4, we reanalyzed the measured photoresponse rate and IQE using this new lifetime value. We note that our model still agrees well with the data, which demonstrate that our analysis remains valid even without a precise knowledge of the exciton recombination time. This is due to the fact that dissociation is the dominant process for the field range studied here. The same holds also for the carrier lifetime, which is dominated by carrier drift at high field. Finally, we point out that the “uncertainty-limited recombination time” is misinterpreted by the reviewer. Clarifications are given in our answer to Question 2 of Reviewer #2.

Figure R3. Time-resolved photoluminescence of an hBN-encapsulated monolayer WSe_2 at room temperature. The black line is a linear fit yielding a recombination time of ~ 300 ps. This figure has been added to the SI.

Figure R4. Photoresponse rate vs in-plane electric field and IQE vs bias voltage (Inset). The model is the same as in the manuscript, except that here the exciton recombination time is $\tau_{r,N} = 300$ ps. For more details, see the caption of Figure 4a of the manuscript.

2. Fig. 4, the authors claim that the dissociation process is the rate-limiting factor for electric field of 10 – 15 V/ μm and the response rate matches that predicted by the tunnel ionization model. These statements are the main finding of this work but appear weakly supported. The model curve in Fig 4, which is the only evidence for the tunnel ionization process, was not directly generated from the model but rather extrapolation using the computed values at larger fields of 18- 24 V/ μm with basically four parameters (Table S1). It is understandable that there are difficulties and limitation in obtaining model curves at small fields, but this indirect approach unavoidably weakens the reliability of their analysis. The authors can consider comparing their measured Stark shift to that predicted by their tunnel ionization model as a possible further support or provide other evidence to back up this main claim.

In fact, this is already what we do: we employ the same theory (Wannier-Mott exciton) to model the measured Stark shift and the measured photoresponse rate. More precisely, by modelling the Stark shift we extract an exciton binding energy of 170 meV. We use this value to calculate (with no fitting parameters) the tunnel ionization rate Γ_{diss} in our photoresponse rate model, which we compare to the measured photoresponse time. This way, we provide a consistent analysis of the Stark shift and photoresponse rate that strongly support our main claim, i.e, exciton tunnel ionization in TMD.

We would also like to stress that contrary to what the referee seems to imply, the tunnel ionization rate Γ_{diss} predicted by our model at low fields is reliable. Indeed, it is well established that Γ_{diss} is dominated by the term $\exp(-F_0/F)$ at low fields (see, for instance, Landau & Lifshitz,

Quantum Mechanics, 1965, p.276). The low field regime corresponds to $F \ll F_0$, which is the case for the field range considered in our experiment.

3. A similar exciton ionization model has been reported recently (New J. Phys. 18 (2016) 073043), which does not rely on an extrapolation at low electric fields and predicts an ionization rate that is more than two-order-of-magnitude larger than the results in this manuscript. Given the direct relevance, this work should be cited and the authors should consider explanations for the above discrepancy.

We are aware of the article mentioned by the reviewer since it was written by one of the co-authors of our manuscript (T. G. Pedersen). In this article, exciton ionization is calculated only for bulk TMDs which have a significantly lower exciton binding energy ($E_B < 50$ meV). The ionization rates are therefore much higher than in the case of monolayer TMD. We added a sentence in the new version of the manuscript to explain this interesting observation.

4. In the analysis, the authors do not consider carrier travel time from the junction to the graphite source/drain. A quick estimate using the mobility of $4 \text{ cm}^2\text{V}^{-1}\text{s}^{-1}$, travel distance of $2 \text{ }\mu\text{m}$, and bias voltage of 1 V yields a drift velocity $v_d = \mu E = 200 \text{ m/s}$ and a corresponding travel time of 10 ns , translating to 0.1 GHz . This is much slower than the extracted response rate of up to 50 GHz . Can the authors comment on this?

The measurement technique that we employ to measure the photoresponse rate, i.e. time-resolved photocurrent, probes the time it takes for photoexcited carriers to escape the photoactive region, which in our case is the p-n junction of length $L = 200 \text{ nm}$. The rest of the WSe_2 channel (the highly P or N regions, which are not photoactive) can be thought of as metallic contacts, just like the graphite flakes or the gold pad connecting them. This technique (and its interpretation) is well described in Sections 5 and 6 of the SI and it has already been employed by our group and others (e.g., PRL 108 087404 '12) to measure photoresponse rates which are not limited by the drift or diffusion of carriers to the contacts.

5. On a similar note to the previous comment, fast response rate of up to 60 GHz is reported at $V_B = 0$ (Fig. 3c) where the collection of photocurrent relies on the diffusion of carriers to the source and drain on μm scales. Is the carrier diffusion fast enough to yield such fast response rate?

Our answer to Question 4 also holds for this question: our measurement technique (and therefore the photoreponse rate we measure) is not sensitive to the transport of charge carriers (diffusion or drift) to the source-drain contacts, but mainly to their escape (via dissociation and drift) out of the photoactive p-n junction.

6. The authors state that “the application of a large in-plane electric field shortens the lifetime of excitons” (Page 5). It is known that an external electric field reduces the overlap of electron and hole wave function and decreases exciton lifetime in the out-of-plane geometry. I would expect similar effect to take place in in-plane geometry. Can the authors clarify this?

First of all, we believe the reviewer meant: “an external electric field reduces the overlap of electron and hole wave function and INCREASES the exciton lifetime (more specifically, its recombination time) in the out-of-plane geometry”.

This effect is predicted to occur in the in-plane geometry as well, but for our experimental field range (~ 20 V/micron), this effect is negligible. Indeed, the radiative recombination rate is proportional to the oscillator strength f of the exciton. According to Scharf et al. (PRB 94 245434 '16), f decreased by $\sim 15\%$ (compared to the zero-field value of f) at 20 V/micron, leading to a corresponding increase in exciton recombination time. As we showed in our answer to Q1, our model and results are not affected by such a small variation in the exciton recombination time. In fact, as we mention in the manuscript, the lifetime of the exciton is predominantly shortened by its “decay into free electrons and hole” (p. 5).

7. What is the typical dark current range in the WSe2 device? In Fig. 3a, it looks like the dark current in that specific gating condition is well below nA range. It will be informative if the authors can include IV characteristics at a few representative gate conditions with and without illumination.

Figure 3a of the manuscript, to which the reviewer refers, shows the photocurrent (PC) as a function of laser power. As explained in the method section, PC is measured using a mechanical chopper and therefore corresponds to the difference between the current with and without illumination ($PC = I_{ON} - I_{OFF}$). Hence, it is not possible to estimate the dark current I_{OFF} from this figure.

Information on the dark current is however presented in Fig S1b of the SI. As the reviewer correctly guessed, the dark current in reversed bias ($V_B > 0$ for $V_{asym} > 0$ and $V_B < 0$ for $V_{asym} < 0$) is well below the nA range (it is in fact below our instrument sensitivity). To answer the reviewer’s request, we show below the IV characteristics in the dark (Fig. R5a) and photocurrent (Fig. R5b) at various gate conditions (V_{asym}). These figures are included in the new version of the SI. We note that since $I_{OFF} \ll I_{ON}$ in most gate and bias conditions, $PC \sim I_{ON}$.

Figure R5. a) Dark current I_{OFF} and b) photocurrent PC vs V_{asym} ($= V_{G1} = -V_{G2}$) and source-drain bias V_B .

8. The authors mention that the in-plane polarizability is much greater than the out-of-plane one in atomically-thin TMDs. This is an interesting observation. Can the author provide a short explanation for it?

This is simply due to the 2D nature of the exciton: the exciton is extended in the in-plane direction (Bohr radius of $\sim 1-2$ nm) and more confined in the out-of-plane direction (layer thickness = 0.65 nm). Since polarizability typically scales (to some power) with the exciton radius (along the field direction), polarizability is expected (and measured) to be larger in in-plane direction.

Reviewers' comments:

Reviewer #1 (Remarks to the Author):

The authors present detailed photocurrent measurements of monolayer WSe₂ encapsulated in hBN. They identify a peak in the photocurrent excitation spectrum as the onset of continuum states that is similar to BSE calculations of the exciton binding energy. They observe a DC Stark shift of the photocurrent excitation spectrum peak corresponding to the A-exciton and from it estimate the exciton polarizability. Using time-resolved photocurrent measurements they estimate an exciton-exciton annihilation rate of 0.05 cm²/s and a EEA-limited exciton lifetime of 10ps. The authors use a dissociation and drift model to describe the in-plane field dependence of the photocurrent and show that dissociation at low fields is consistent with tunnel ionization.

These results are of broad interest to the multidisciplinary field of researchers studying 2D materials. However, I think they need to be put in perspective with recent work in the field, which reduces the novelty of this work. There are also several points in the manuscript that need to be clarified and claims that need to be better supported. Therefore, I have to recommend that the manuscript be rejected at this time. After revision, it may be more suitable for publication in a journal with less strict novelty and immediacy requirements.

Comments:

1. The novelty claims on this manuscript depend on it being the first to address the underlying physical origin of exciton dissociation on 2D TMDs. However, a recent study of exciton fission by Stienhoff et al. Nat Commun v8 p1166 2017 entitled "Exciton fission in monolayer transition metal dichalcogenide semiconductors" has been ignored. The authors should discuss this important study and compare their results to those in that work.

2. Graphite is used as the electrical contact. The authors use Ref 30 to justify this choice, claiming it makes a high quality ambipolar contact.

a) Ref 30 uses graphene as the contact, not multilayer graphite. The two cannot be assumed to behave equivalently

b) Space charge build up may contribute to the reported sublinearity excitation dependence, where charge build up at electrodes produces a field that reduces charge flow and thereby reduces current. It has not established that ohmic contact is made, which is needed to avoid this effect.

c) It is known that dielectric screening will also occur near graphene, and presumably graphite, see e.g. Nat Commun 8 15251. This can lead to >100meV reduction in the exciton binding energy and easier exciton ionization.

3. The authors claim that the ionization rate is highest in the middle of the gap between electrodes based on finite element analysis of the electric field. This assumes that the ionization rate depends only on field. However, Chernkov et al PRL 115 126802 '15 show that the exciton binding energy is reduced due to charge injection from electric fields, which would in turn increase the ionization rate for a given field strength. Therefore, it cannot be assumed that the ionization rate is highest in the undoped regions.

4. The authors assume that bandgap renormalization can be neglected over their experimental excitation density range. However, recent theoretical (Nat Commun v8 p1166 2017) and experimental work (DOI:10.1021/acsnano.7b06885) shows that the electronic bandgap can be reduced by ~100meV for $N \sim 1e12$ due to dipolar screening of Coulomb interactions by excitons.

(a) This contradicts the claim that bandgap renormalization only contributes for $N > 1E13$.

(b) Even if EEA is found to dominate, both dipolar and charge carrier screening effects will contribute to the dependence on excitation density. The corresponding reduction in bandgap should lead to lower fields necessary to ionize carriers. The observed sub-linear excitation density dependence implies that EEA dominates charge carrier creation, where it rapidly reduces the exciton population available to be ionized. Bandgap renormalization may be a small correction.

This underlying physics should be discussed in the manuscript.

5. The exciton dynamics are dominated by EEA, which limits the lifetime to ~ 10 ps. However, the model applied to describe exciton dissociation assumes an exciton lifetime of 1ns. Further, the measurements show a low-fluence exciton lifetime of 300ps. Please clarify these apparent discrepancies.

6. While uncertainty is given for the estimated polarizability, and this uncertainty has been propagated into the theoretical binding energy, no experimental uncertainty in the experimental binding energy has been given. For example, a binding energy of 140meV (i.e. $1.87\text{eV} - 1.73\text{eV}$) seems to have been experimentally determined yet the BSE calculation arrives at 170meV. We can only evaluate their agreement based on uncertainty. Figure S2 shows difference energy separations between the peak near 1.7eV (exciton) and the step-increase near 1.9eV (bandgap) implying the binding energy varies among samples.

7. Note that Ref 39-41 also discuss EEA and should be referenced as such in the introduction.

Reviewer #2 (Remarks to the Author):

I am satisfied with the revisions and recommend publication of the revised manuscript.

Reviewer #3 (Remarks to the Author):

The authors have properly addressed my previous comments. Their detailed clarification and justification are appreciated. I would recommend the revised manuscript for publication in Nature Communication.

We thank Reviewers #2 and 3 for supporting the publication of our revised manuscript. We also acknowledge the constructive comments of Reviewer #1 which we address below.

Please note that reviewers' comments are in blue font and that all changes made to the manuscript and Supplementary Information (SI) are highlighted in green.

Reviewer #1 (Remarks to the Author):

The authors present detailed photocurrent measurements of monolayer WSe₂ encapsulated in hBN. They identify a peak in the photocurrent excitation spectrum as the onset of continuum states that is similar to BSE calculations of the exciton binding energy. They observe a DC Stark shift of the photocurrent excitation spectrum peak corresponding to the A-exciton and from it estimate the exciton polarizability. Using time-resolved photocurrent measurements they estimate an exciton-exciton annihilation rate of 0.05 cm²/s and a EEA-limited exciton lifetime of 10ps. The authors use a dissociation and drift model to describe the in-plane field dependence of the photocurrent and show that dissociation at low fields is consistent with tunnel ionization.

These results are of broad interest to the multidisciplinary field of researchers studying 2D materials. However, I think they need to be put in perspective with recent work in the field, which reduces the novelty of this work. There are also several points in the manuscript that need to be clarified and claims that need to be better supported. Therefore, I have to recommend that the manuscript be rejected at this time. After revision, it may be more suitable for publication in a journal with less strict novelty and immediacy requirements.

Comments:

1. The novelty claims on this manuscript depend on it being the first to address the underlying physical origin of exciton dissociation on 2D TMDs. However, a recent study of exciton fission by Stienhoff et al. Nat Commun v8 p1166 2017 entitled "Exciton fission in monolayer transition metal dichalcogenide semiconductors" has been ignored. The authors should discuss this important study and compare their results to those in that work.

We thank the reviewer for his/her constructive feedback; however, we strongly disagree with the reviewer's statement about the lack of novelty of our work. The study of Stienhoff et al. mentioned by the reviewer is (i) a **theoretical** analysis of the thermodynamic of excitons and free carriers (ii) which does not consider the effect of external electric field on exciton dissociation and (iii) was published three weeks after we submitted our manuscript to Nature Communications. In our manuscript, we quantitatively and experimentally address for the first time the mechanism that leads to dissociation of 2D excitons in TMDs: tunnel ionization. This mechanism is not considered at all by Stienhoff et al.

2. Graphite is used as the electrical contact. The authors use Ref 30 to justify this choice, claiming it makes a high quality ambipolar contact.

a) Ref 30 uses graphene as the contact, not multilayer graphite. The two cannot be assumed to behave equivalently

First of all, we do not claim that graphite makes a "high quality ambipolar contact", but rather that it "serves as ambipolar electrical contact". Secondly, we do not simply assume this claim. In the Supplementary Information (Figure S1a), we show that our electrical device can operate in both

electron- (NN) and hole-doped (PP) regimes. In Figure S1b, we show the device has an ideal ohmic behavior in the PP configuration, while it still displays residual rectification in the NN configuration. This rectification is however much smaller than the one created by the PN or NP junction (see Figure S1b).

b) Space charge build up may contribute to the reported sublinearity excitation dependence, where charge build up at electrodes produces a field that reduces charge flow and thereby reduces current. It has not established that ohmic contact is made, which is needed to avoid this effect.

In addition to the explanation given in the previous answer, at least two other experimental observations allow us to exclude space charge build up at electrodes and to confirm exciton-exciton annihilation as the origin of the sublinear power dependence. First, space charge build up at the WSe₂/graphite contact would lead to a potential drop which, upon local illumination (of the contact), would lead to photocurrent (PC) generation. However, photocurrent maps measured over the entire sample (e.g. Fig. 1c) show no detectable photocurrent at the WSe₂/graphite interface. The photocurrent map only shows clearly observable photocurrent at the p-n junction.

Secondly, the sublinear power dependence of the photocurrent we report is similar to the one we observe when we measure the photoluminescence (PL) of WSe₂ (see Fig. R1 below). According to our model described in section 5 of the SI, this sublinear PL behavior - which has been reported several times (e.g. Mouri, Y. et al., Phys. Rev. B 90, 155449, 2014) - should follow $PL \propto \ln(1 + \gamma\tau N_0)$. By fitting the PL data with this equation (red line in Fig. R1), we obtain $1/\gamma\tau \sim 5e11 \text{ cm}^{-2}$, which is similar to the values reported in our manuscript. Since the origin of PL sublinearity cannot be attributed to space charge build up but is rather consistent with exciton-exciton annihilation, we conclude together with the spatial photocurrent images (as discussed above) that the latter effect is indeed responsible for the observed PC sublinearity.

Figure R1. Time-integrated photoluminescence vs laser power. Data points were measured using the same laser as the PC data shown in Fig. 3a, but with $\lambda = 532 \text{ nm}$. The red line is a fit to the data using $PL \propto \ln(1 + \gamma\tau N_0)$, where N_0 is defined in equation S14.

c) It is known that dielectric screening will also occur near graphene, and presumably graphite, see e.g. Nat Commun 8 15251. This can lead to $>100\text{meV}$ reduction in the exciton binding energy and easier exciton ionization.

We agree that the dielectric screening created by the graphite flakes will likely reduced the exciton binding energy in the adjacent WSe₂ flake. However, the graphite flakes are far (>5 micron) from the photoactive area of interest (i.e., the p-n junction in WSe₂). In addition, as mentioned in the previous answer, when performing PC maps, no PC is observed at the WSe₂/graphite junction. This is likely due to the lack of significant potential drop at this interface, which is required to dissociate the excitons and sweep the free carriers.

3. The authors claim that the ionization rate is highest in the middle of the gap between electrodes based on finite element analysis of the electric field. This assumes that the ionization rate depends only on field. However, Chernkov et al PRL 115 126802 '15 show that the exciton binding energy is reduced due to charge injection from electric fields, which would in turn increase the ionization rate for a given field strength. Therefore, it cannot be assumed that the ionization rate is highest in the undoped regions.

As we mentioned in our previous round of answers to the reviewers (Reviewer #1, Question #3), “we also performed more detailed calculations (not shown) of E_B in doped WSe₂ using the Wannier-Mott model (presented in Section 3 of the SI) which confirm that the ionization rate is largest in the undoped, inter-gate region”. We now provide the results of these calculations.

Figure R2a, below, shows the calculated exciton binding energy E_B as a function of charge carrier density n . As the reviewer correctly points out, E_B is reduced at large n . To verify how this effect affects the ionization rate in our device, we calculate the dominant ionization term $\exp(E_B/dF)$ across the p-n junction, where d is the exciton radius ($d \sim 1\text{nm}$) and F is the in-plane electric field. The results, shown in Fig. R2b, clearly demonstrate that the ionization rate is highest in the undoped region. This confirms the claim we made in the previous round of answers: “The exciton ionization rate is always higher in this region where the exciton binding energy is unaffected by Coulomb screening. This means that the exciton binding energy relevant to our analysis is the one corresponding to the undoped WSe₂.”

Figure R2. **a)** Calculated exciton binding energy E_B vs charge carrier density n . **b)** Exponential ionization term $\exp(E_B/dF)$ vs position x . The middle of the split gate is located at $x = 0$. Both $F(x)$ and $n(x)$ were taken from the calculations shown in Fig. S4b.

4. The authors assume that bandgap renormalization can be neglected over their experimental excitation density range. However, recent theoretical (Nat Commun v8 p1166 2017) and experimental work (DOI:10.1021/acsnano.7b06885) shows that the electronic bandgap can be reduced by $\sim 100\text{meV}$ for $N \sim 1\text{e}12$ due to dipolar screening of Coulomb interactions by excitons. (a) This contradicts the claim that bandgap renormalization only contributes for $N > 1\text{E}13$. (b) Even if EEA is found to dominate, both dipolar and charge carrier screening effects will contribute to the dependence on excitation density. The corresponding reduction in bandgap should lead to lower fields necessary to ionize carriers. The observed sub-linear excitation density dependence implies that EEA dominates charge carrier creation, where it rapidly reduces the exciton population available to be ionized. Bandgap renormalization may be a small correction. This underlying physics should be discussed in the manuscript.

We agree with the reviewer's comments: Bandgap renormalization (BR) can occur for moderate excitation intensities, below $N = 1\text{e}13\text{cm}^{-2}$, but the sublinear power dependence we observed imply that the contribution of BR is small compared to exciton-exciton annihilation (EEA). As Stienhoff et al. (Nat Commun v8 p1166 2017) point out, "screening in a correlated many-particle system near the Mott transition is an intricate problem" and is beyond the scope of our manuscript. We added a sentence in the new version of the manuscript to emphasize this point.

5. The exciton dynamics are dominated by EEA, which limits the lifetime to $\sim 10\text{ps}$. However, the model applied to describe exciton dissociation assumes an exciton lifetime of 1ns . Further, the measurements show a low-fluence exciton lifetime of 300ps . Please clarify these apparent discrepancies.

In our model (presented in Sections 5 and 6 of the SI), the power dependent EEA lifetime is taken into account by a separated term (γN^2), which allows us to untangle its contribution from the "intrinsic" (i.e. power independent) response time τ of the device. This term depends on both the exciton recombination and ionization times. We estimated the exciton recombination time using two approaches. First, we fitted the data in Fig 4a of our manuscript with the model described in the manuscript and found $\sim 1\text{ ns}$. We note the uncertainty on this value is large and difficult to calculate due to the lack of data at low electric field. We also measured the exciton recombination time on a similar sample through time-resolved PL measurements and extracted an exciton recombination time of $\sim 300\text{ ps}$. As we explained in our previous round of answers to the reviewers, this value is in reasonable agreement with the previous estimation ($\sim 1\text{ ns}$).

6. While uncertainty is given for the estimated polarizability, and this uncertainty has been propagated into the theoretical binding energy, no experimental uncertainty in the experimental binding energy has been given. For example, a binding energy of 140meV (i.e. $1.87\text{eV} - 1.73\text{eV}$) seems to have been experimentally determined yet the BSE calculation arrives at 170meV . We can only evaluate their agreement based on uncertainty. Figure S2 shows difference energy separations between the peak near 1.7eV (exciton) and the step-increase near 1.9eV (bandgap) implying the binding energy varies among samples.

The experimental binding energy is estimated by two approaches, both of which depend to some extent on theoretical models. The first method, to which the reviewer refers, consists in estimating E_B based on the measured photocurrent spectrum. We indeed observe an exciton peak at 1.73 eV

and a step-like increase at 1.87 eV. However, this step-like increase **cannot** be attributed directly to the position of free particle bandgap (as the reviewer seems to imply). To determine the position of the bandgap, we compare the calculations of the BSE and Wannier-Mott models to our measurements (Fig. 2b of the manuscript). We find that theory and measurements agree with a bandgap of 1.9 eV. Hence, we find $E_B = 1.9 - 1.73 = 0.17$ eV. Since this value partly relies on a theoretical model, its uncertainty can hardly be defined.

The second approach to estimate E_B is to measure the exciton polarizability and compare it to the theoretical E_B . Since the value of exciton polarizability does not depend on theory, its uncertainty can be obtained and propagated into theoretical uncertainty on E_B .

7. Note that Ref 39-41 also discuss EEA and should be referenced as such in the introduction. We thank the reviewer for his/her observation. The references have been added in the introduction.

Reviewer #2 (Remarks to the Author):

I am satisfied with the revisions and recommend publication of the revised manuscript. We thank the reviewer for his/her recommendation.

Reviewer #3 (Remarks to the Author):

The authors have properly addressed my previous comments. Their detailed clarification and justification are appreciated. I would recommend the revised manuscript for publication in Nature Communication.

We thank the reviewer for recommending the publication of our manuscript.

REVIEWERS' COMMENTS:

Reviewer #1 (Remarks to the Author):

The authors have adequately addressed all concerns.

We thank Reviewer #1 for his/her positive comment on our revised manuscript.

Reviewer #1 (Remarks to the Author):

The authors have adequately addressed all concerns.